# External Ventricular Drains: Development and Evaluation of a Nursing Clinical Practice Guideline

**Tainara Wink Vieira** [1,2,*] , **Victória Tiyoko Moraes Sakamoto** [2] , **Bárbara Rodrigues Araujo** [1] , **Daiane Dal Pai** [2] , **Carine Raquel Blatt** [1] **and Rita Catalina Aquino Caregnato** [1]

1  Nursing Department, Federal University of Health Sciences of Porto Alegre (UFCSPA),
   Porto Alegre 90050-170, Brazil
2  Medical-Surgical Nursing Department, Federal University of Rio Grande do Sul (UFRGS),
   Porto Alegre 90040-060, Brazil
*  Correspondence: tainara.winkv@gmail.com

**Abstract:** External ventricular drains (EVDs) are common in intensive care for neurocritical patients affected by different illnesses. Nurses play an essential role to ensure safe care, and guidelines are tools to implement evidence-based care. Thus, the aim of this study was to develop and evaluate the quality of a clinical guideline for critically ill patients with EVDs. Methodological research was conducted. The guideline development was based on a scoping review about nursing care to patients with EVDs. The guideline evaluation occurred in two phases: evaluation of its methodological rigor, with application of the Appraisal of Guidelines Research and Evaluation II to four experts on guidelines evaluation; and the Delphi technique, with a panel of nine specialists in neurocritical care, performed in two rounds. Data were analyzed by descriptive statistics and content validity ratio. In the first phase of the evaluation, three domains did not reach consensus, being reformulated. The second phase was conducted in two rounds, with nine and eight participants respectively, with 13 recommendations being reformulated and reassessed between rounds, inclusion of an EVD weaning category, and two flowcharts on patient's transport and mobility. Therefore, the guideline can be incorporated into nursing care practices. Further studies are necessary to assess its impact on clinical practice.

**Keywords:** cerebrospinal fluid shunts; practice guideline; validation study; nursing; critical care; evidence-based practice

## 1. Introduction

The external ventricular drain (EVD) is a technology frequently used in patients submitted to neurosurgical procedures [1,2], being indicated as a treatment mainly in cases of hydrocephalus and intracranial hypertension (ICH) secondary to intracranial hemorrhages, head trauma, and infections affecting the central nervous system [1–3]. The EVD is a diagnostic and therapeutic tool [4,5] for allowing both drainage of the cerebrospinal fluid (CSF) and measurement of the intracranial pressure (ICP), enabling intensive neurological monitoring of patients affected by such illnesses [3].

However, despite its routine use in intensive care settings, there is a lack of studies with high levels of evidence and recommendations about the management of the device [6,7]. It is also known that the benefits of the EVD are minimized by the risk of complications associated with its use [4,6,8]. Among these are infection [9], drain obstruction, excess fluid drainage, and accidental removal of the EVD system [1,10]. These complications may cause hemorrhages, ventricular complications, and need for neurosurgical intervention [1,10].

Guidelines related to the theme are still incipient in literature [11,12], despite the description of good results related to decreased infection rates in hospitals that implemented prevention bundles and guidelines for insertion and maintenance of the EVD [5,6,9,13–21].

At the time of writing this paper, no article was found on the implementation of guidelines related to the topic in Brazilian hospitals, except for descriptions of the impact of educational activities on staff knowledge [14] and infection rates [15].

A meta-analysis published in 2018 included eight studies which analyzed infection rates before and after implementation of standardized EVD protocols [16]. The mean pre-protocol infection rates compared to post-protocol were 16.11% versus 4.67%, although relative risk of infection showed high heterogeneity and a substantial risk of publication bias. Despite this, the authors indicate a significant positive association between pre-protocol infection rates and the percentage reduction in infection rates after the implementation of a protocol, suggesting that institutions may benefit from its implementation.

Additionally, authors share their successful experience implementing the EVD insertion and handling protocol in their institution, zeroing infection within 15 months of protocol initiation, from a previous infection rate of 12.4% [16]. Other studies [9,13,16–19] also report satisfactory results related to infection reduction and implementation of protocols, despite the high variability among the assessed protocols.

For safe care, evidence-based practice (EBP) is mandatory. EBP corresponds to the provision of care based on the best available evidence according to the local reality [20], and guidelines are key for standardization of actions and help to implement evidence-based care. Clinical guidelines aim at providing adequate care in an efficient manner, ensuring more benefits than harm [20]. In addition, they facilitate decision-making and provide greater safety to the team, since they reduce the variability of actions [21,22].

Therefore, the aim of this study was to develop and evaluate the quality of a clinical practice guideline for critical patients with external ventricular drains. At the beginning of this research, there were not any published guidelines on the subject in Portuguese and adapted to Brazilian reality, justifying the need for a new tool.

## 2. Methods

This is a methodological study on the development of the "Nursing clinical practice guideline for critical patients with EVD", consisting of two phases, namely: (1) development of the "Clinical practice guideline for critical patients with EVD", and (2) evaluation of the guideline's quality.

The study was conducted in accordance with the Declaration of Helsinki and approved by the Ethics Committee of the Federal University of Health Sciences of Porto Alegre (protocol code 4.096.987 and 18 June 2020), located in Porto Alegre, RS, Brazil.

### 2.1. Development of the "Clinical Practice Guideline for Critical Patients with EVD"

This consisted in three sequential steps:

(1) scoping review on nursing care in patients with EVD; indispensable
(2) quality assessment of the studies included in the scoping review using the Grading of Recommendations, Assessment, Development, and Evaluation (GRADE) approach;
(3) development of the guideline following the framework of Pimenta et al. [21].

Data from the first and second steps are already published and available for consultation [23] and therefore will not be detailed in this paper. The third step considered the recommendations of Pimenta et al. [21] to structure clinical guidelines, and the Appraisal of Guidelines Research and Evaluation II (AGREE II) [24] recommendations for rigorous methodological development. It included these topics: source (institution responsible for the guideline development), aims, research team, conflicts of interest, recommendations for practice, external review and update, flowcharts, outcomes assessment, professionals and users evaluation, limitations plan implementation.

### 2.2. Quality Assessment of the "Clinical Practice Guideline for Critical Patients with EVD"

This consisted of two steps, namely:

(4) assessment of the methodological rigor of the guideline, with the application of AGREE II for experts on guidelines assessment;

(5)    Delphi study to evaluate experts' opinion on the recommendations of the guideline.

2.2.1. Assessment of the Methodological Rigor of the Guideline

The first step of data collection was conducted online in the period from July to August 2020. Convenience sample, following the snowball method, was adopted. The sample consisted of four participants, as suggested by AGREE II. Inclusion criteria were experience in guideline evaluation using the proposed tool. Seven participants were invited, and three were excluded for not returning their answers by the deadline date.

The AGREE II tool was used, which addresses variability in the quality of clinical guidelines, assessing methodological rigor and transparency in their development. AGREE II is composed of 23 items assessed through a seven-point Likert scale, from 1 (strongly disagree) to 7 (strongly agree), with room for comments. The items are distributed over six domains (1—scope and purpose, 2—stakeholder involvement, 3—development rigor, 4—clarity of presentation, 5—applicability, 6—editorial independence) [24].

AGREE II was submitted in Google Form format via electronic mail. The form also included closed questions to characterize the sample, such as: previous evaluation of guidelines using the AGREE II, number of guidelines evaluated, age, gender, job, professional background, and expertise area. Along with the form, participants received an explanatory text, the informed consent form (ICF), and the complete "Clinical practice guideline for critical patients with EVD", with a watermark of unauthorized version for disclosure.

Data were tabulated in Microsoft Excel 2016 (Microsoft Corporation, Porto Alegre, Brazil) and analyzed according to the scoring system proposed by AGREE II. A desirable concordance rate of 80% per domain was considered.

2.2.2. Delphi Study

The second step was a Delphi study. The Delphi technique is a reliable and systematic way to obtain consensus among a group of experts, from the application of well-defined questionnaires in two rounds and feedback. It is often used in guideline development worldwide, since it relies on expert knowledge and experience to negotiate a shared reality and to co-construct knowledge [25]. It also assures anonymous answers to avoid pressure and conformity to dominant view, and it was particularly important during the 2019 Coronavirus Disease pandemic, considering that some participants were frontline professionals in intensive care units and could not have synchronous meetings for the study's purposes.

Data collection occurred from July 2020 to January 2021. The deadline for responses was extended given the pandemic. This phase was also composed of a convenience snowball sample. The initial sample considered 10 participants. An invitation to participate was sent to 25 subjects, and nine accepted to participate and answered questionnaires on time in the first round. In the second round, eight of them returned questionnaires by deadline. Physicians and nurses with a minimum experience of two years in the ICU with emphasis in neurocritical patients and specialization in Intensive Care or related areas, Neurology and/or Neurosurgery, were included upon acceptance of the ICF. The 16 participants who did not answer the questionnaires within the agreed time frame were excluded.

Data collection for this step also occurred online by sending questionnaires in Google Form format by electronic mail. Participants also received with the form an explanatory text, the informed consent form, and the full guideline, as in the previous step. Thus, two rounds were conducted.

The questionnaire for the first round included the following items: (a) explanatory text about research data; (b) closed questions to characterize the sample; and (c) recommendations of the guideline, each evaluated with a four-point Likert scale ranging from 1 (strongly disagree) to 4 (strongly agree), and open question for comments. After the end of the first round, data were analyzed and used for the formulation of the second questionnaire sent to the participants.

The analysis in the second step was performed by descriptive statistics, analysis of the experts' suggestions, and the content validity ratio (CVR), proposed by Lawshe (1975) [26], which is a linear transformation of a proportional level evaluating how many specialists consider an item "essential"—in this case, "strongly agree".

Lawshe's (1975) [26] table of critical CVR values was adopted as a reference, considering Critical CVR = 0.77 in the first round, with nine specialists. Thus, the second questionnaire was composed of 13 recommendations for evaluation, which were reformulated based on the results obtained in the previous round (CVR ≤ 0.77 and/or descriptive suggestions from the participants). Data from the second round were analyzed considering Critical CVR = 0.75, as the final panel was composed of eight experts. Figure 1 below illustrates the whole process.

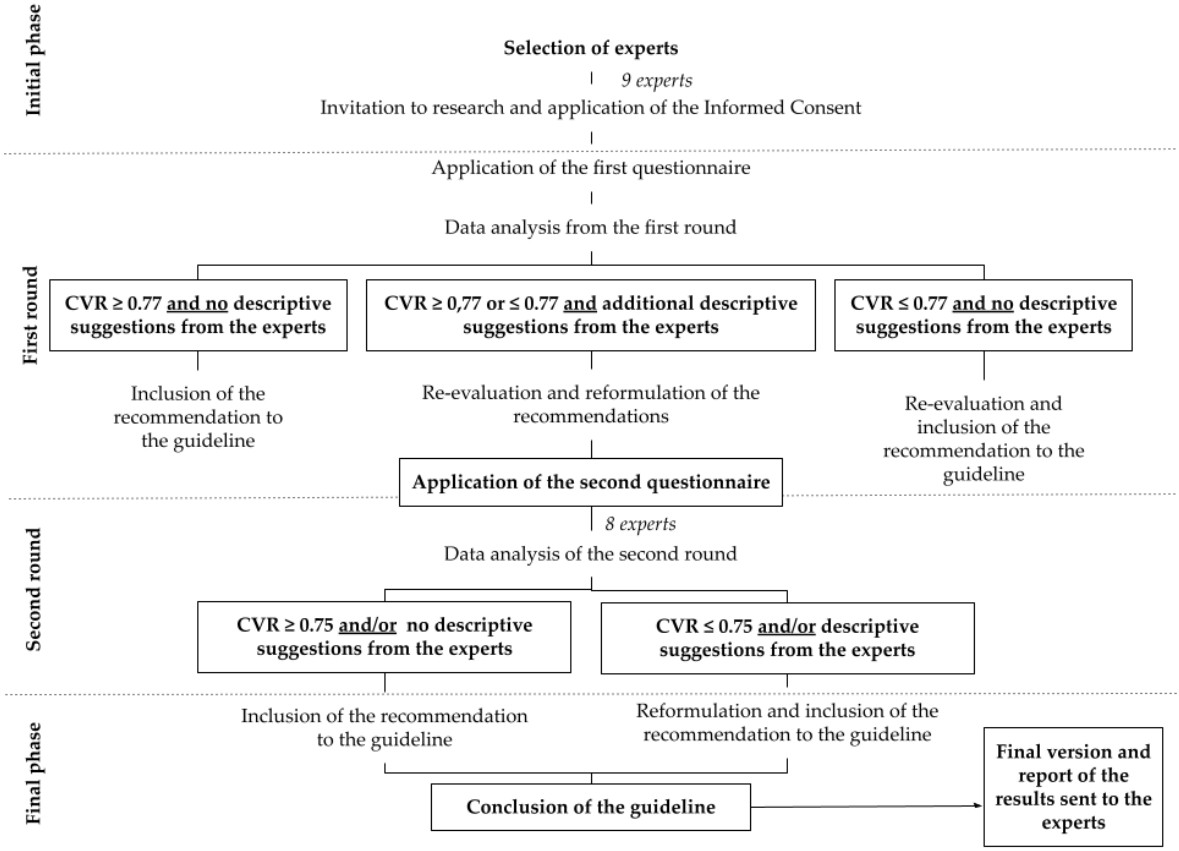

**Figure 1.** Process flowchart of the Delphi study.

## 3. Results

### 3.1. Development of the Guideline

The final version of the guideline consisted of the following sections: presentation of the EVD; contextualization for the use of the guideline; methods for its development; algorithms for initial assessment of the neurocritical patient with EVD, bed bath of the patient with EVD, transport and mobilization of the patient with EVD; ten recommendations for nursing care; instructions for performing procedures (dressing the drain site and emptying the collection bag); plans for evaluation, implementation, and outcome indicators of the guideline; facilitating aspects and limitations for its use; and proposal of a bedside care plan in the ICU.

The first version of the guideline consisted of nine recommendations and did not include algorithms for transport and mobilization of the patient with EVD and a recommendation to wean off EVD. These modifications were made through the evaluation process, which is described in the following two subsections.

### 3.2. Assessment of the Methodological Rigor of the Guideline

The evaluation of the guideline in this step was performed by four health professionals, named P1, P2, P3, and P4. Three were pharmacists with specialization (1), master's (1) and doctoral (1) degrees in Pharmaceutical Sciences and Hepatology, and one undergraduate student in Pharmacy. All participants had experience in the evaluation of guidelines with the AGREE II instrument, ranging from 2 to 90 guidelines evaluated per participant.

Scoring was given by domains. Domains "scope and purpose" (93.05%), "clarity of presentation" (83.33%), and "editorial independence" (85.41%) scored above 80%, and did not have changes. Domains "stakeholder involvement" (66.66%), "rigor of development" (68.75%), and "applicability" (75.00%) obtained scores lower than 80%, below the minimum value of adequacy proposed by authors; therefore, these had to be re-evaluated.

Regarding the domain "stakeholder involvement", which scored the lowest (66.66%), lower scores were assigned to the item related to knowledge of opinions and preferences of the target population, suggesting that in the next updates of the guideline, such aspects should be considered. In addition, there was a discrepancy in scores regarding the composition of the guideline development team.

The domain "rigor of development" refers to the method used to search for the evidence on which the guideline is based, and the participants suggested a more detailed and transparent description of the scoping review. The participants' suggestions were included in the final version of the guideline.

The domain "clarity of presentation" showed lower scores on the item regarding the approach to different treatment options. The guideline is directed to specific EVD care with emphasis on nursing care, so that it is not within its scope to discuss different treatment strategies and criteria for their use and other technologies, since this is a medical decision exclusively. Thus, no changes were made to the guideline in this aspect.

The "applicability" domain presented medium scores in the item related to potential resource implications of applying the recommendations. The domain "editorial independence" showed an overall score of 85%, with a discrepancy of scores only in the item related to the influence of the opinion of the funding agency on the content of the guideline. However, there were not any comments from participants related to this item, so no modifications were made in this topic of the guideline.

Two participants recommended the guideline use, and the other two recommended its use with modifications. The proposed changes regarding the domain "rigor of development" were incorporated into the guideline. The other considerations were justified and were not changed in the original guideline.

### 3.3. Delphi Study

The expert panel was composed of nine participants in the first round, five nurses and four physicians. Eight participants attended the second round (physicians = 3, nurses = 5), corresponding to an abstention rate of 11.1% between rounds.

Most of the participants were female ($n = 5$; 55.50%), with a mean age of 37.1 years, and a mean experience in Intensive Care and/or Neurology/Neurosurgery of eight years. Regarding their degrees, six (66.60%) had master's degrees and specialization. The areas of specialization were Intensive Care ($n = 4$, 44%), Neurology ($n = 3$, 33%), and Neurosurgery ($n = 2$, 22%). Four participants (44.40%) stated that they had scientific production on the subject (EVD) in the last 10 years.

Twenty-one recommendations from the first version of the guideline were analyzed by the experts in the first phase of the panel (Table 1). Of these, six reached full consensus (CVR = 1.00) or partial consensus (CVR = 0.77) and were not reformulated. Of the remaining recommendations, ten showed CVR below the reference value (CVR ≥ 0.77) adopted by the authors and were reformulated. Two recommendations presented CVR = 0.55, but were not reformulated because there were no suggestions from the experts. Three recommendations presented CVR = 1.00, but were reformulated based on the experts' suggestions.

**Table 1.** Final recommendations after round 1 and 2 of the evaluation process.

| | Final recommendations | CVR * | |
|---|---|---|---|
| | | Round 1 | Round 2 |
| | Non-reformulated recommendations after two rounds, with no additional suggestions from participants. | | |
| ■ | Dressing should be performed with 0.9% saline and 0.5% alcoholic chlorhexidine every 24 h, or earlier if necessary, assessing the aspect of the operative wound and inspecting catheter insertion, covering with sterile gauze, wrapping with bandages. Dressing change is usually performed by nurses, and if there is any change with the insertion of catheter, the neurosurgery team must be notified [13]. | 0.77 | NA |
| ■ | Dressing changes, when using impermeable film, should be performed weekly, or earlier if the dressing is in unacceptable condition, such as peeling off or dirty, in order to minimize direct contact of the catheter with the external environment [13,19,27]. The transparent dressing provides better visualization of the catheter insertion site and allows monitoring of the catheter insertion site [27]. | 0.55 | NA |
| ■ | System handling should be kept to a minimum to ensure that infection risks are minimized. Procedures touching EVD components, such as the sampling port or drainage bag or dressing changes, for example, must be sterile [28]. | 1.00 | NA |
| ■ | The drainage bag should be emptied when it reaches 2/3 or 3/4 of its volume capacity, as if it is too full, it can become heavy and could alter or even stop the function of the system and the drainage of CSF [28]. | 0.55 | NA |
| ■ | If the catheter has been pulled, it must not be repositioned, or even aspirate or administer solutions when it is obstructed. The neurosurgery team must be called whenever there are any changes, due to the high risk of infection and complications [29]. | 1.00 | NA |
| ■ | The nursing team should be aware of any changes in the color of the drained CSF, which may change according to the patient's clinical condition. If blood is present in the CSF, it may be indicative of cerebral hemorrhage; if there is a cloudy or sedimented appearance it may be indicative of infection. If any atypical staining is noted, the neurosurgery team must be called [29]. | 1.00 | NA |
| ■ | CSF collection should be performed only when infection is suspected. Routine CSF collection is not recommended [13]. | 1.00 | NA |
| ■ | 0.5% alcoholic chlorhexidine is recommended for disinfecting the sample port from the EVD [30,31]. | 1.00 | NA |
| | Reformulated recommendations after round 1 and CVR < 0.77 or additional suggestions from participants. | | |
| ■ | The bedside position of the patient should be kept at a 30° angle, head in neutral position and aligned to the cervical spine. The aim is to facilitate venous return, reducing intracranial pressure without interfering in the EVD drainage system [29,32]. | 0.55 | 1.00 |
| ■ | Clamping the EVD system is safe as long as ICP < 20 mmHg and cerebral perfusion pressure (CPP) remains between 60 and 70 mmHg [30]. The system should be kept clamped for as short a time as possible, then unclamped and the entire system checked. It is important to continuously monitor ICP, including during procedures, examinations, and patient transport [29,31]. | 0.11 | 0.75 |
| ■ | The EVD system must be reviewed in order to avoid changes in drainage back pressure and to provide the correct measurement of intracranial pressure. The erroneous leveling of the system can cause alterations in the liquoric drainage, and possible complications in this case are intracranial hypertension, ventricular collapse, and subdural hematoma, among others [7,30]. | 0.55 | 0.50 |

**Table 1.** *Cont.*

| | | | |
|---|---|---|---|
| ■ | The height of the CSF drainage level corresponds to a horizontal line from the Monro foramen—at the level of the external acoustic meatus (EAM)—to the level of back pressure prescribed by the neurosurgeon, usually between 10 and 20 cmH2O. An open EVD at 10 cm above the EAM means that if the ICP is 0 to 10 cmH2O, there will be no drainage [26,28,29]. However, if the ICP is greater than 10 cmH2O, there will be drainage [33]. It is not possible to estimate ICP based on CSF drainage volume, as this is a measurement obtained by other methods [34]. | 0.11 | 0.75 |
| ■ | Check the appearance of dressing every 6 h for moisture indicative of CSF leakage or for signs of inflammation at the catheter insertion. A sterile dressing should be applied to the insertion site when the catheter is properly positioned and there are no signs of infection or CSF leakage, and should be performed by experienced, specialist nurses. It must remain occlusive and dry continuously, covering the insertion. The frequency of dressing change should follow the standardized format of each institution, according to the coverage used and what is indicated by the manufacturer, but it should also be changed whenever it is dirty/wet and should be handled as little as possible [27]. | 0.33 | 1.00 |
| ■ | Rigorous evaluations of consciousness level, using the Glasgow Coma Scale, should be performed, especially in cases of confused or cognitively impaired patients, to ensure that the catheter remains adequately secured and is not pulled out or removed accidentally due to some period of psychomotor agitation. It may also detect early neurological deterioration and signs of sensory impairment due to excessive CSF drainage. In addition, pain should be assessed concurrently with the Glasgow Coma Scale, using standardized scales appropriate for the patient, since pain can be one of the causes for agitation [13]. | 0.55 | 1.00 |
| ■ | The volume of CSF drained in ml every 6 h shift must be recorded, calculating the 24 h total. Drainage volume depends on numerous variables (for example: individual physiological production–450 to 700 mL/day, underlying disease, communicating or non-communicating hydrocephalus, bleeding, leveling of the system, etc.). In some situations, increased volume is a reflection of the underlying disease, and this physiological response is important to maintain adequate or near normal CPP. However, if there is a significant change in drainage volume in a short period of time, the entire system should be reviewed, as should the positioning of the headboard. If nothing is identified, neurosurgical evaluation should be requested. If an error in the manipulation of the system is identified, the team should be provided with guidance in real time [29]. | 0.55 | 0.50 |
| ■ | When the EVD system is open, the transducer may not represent the ICP waveform correctly. To measure ICP with the EVD system, it is necessary to clamp the system every hour as briefly as possible until the P1, P2, and P3 waves are formed, which indicate more accurate ICP, and then to unclamp the system [30]. | 0.33 | 1.00 |
| ■ | When drugs are introduced by the neurosurgery team through the EVD catheter, such as tissue plasminogen activator for intraventricular hemorrhage or antibiotics for ventriculitis, for example, the system should be closed for 1 h after administration so that it is not drained along with excess CSF, as long as there is no significant change in ICP and CPP [30]. | 0.33 | 0.75 |
| ■ | It is recommended that samples be collected from the proximal port, using aseptic technique. The manipulation of the system itself already carries a higher risk of infection, and routine collection is not recommended, so that the material collected from the proximal port with proper care is more reliable [13]. This procedure is performed by the neurosurgery team or by trained nurses, when the neurosurgery team agrees. In addition, samples should not be collected from the collection bag due to rapid degradation of CSF components [30]. | 0.11 | 0.75 |

**Table 1.** *Cont.*

| | | | |
|---|---|---|---|
| ■ | It is important to observe whether the system's dropper flow is properly positioned. If there is minimal or no drainage, the system should be checked for kinks, obstructions, or any clogging. You can also use the system permeability technique to ensure that there is no obstruction by carefully bringing the system below the level to check for CSF dripping. Reduced drainage can cause remodeling of the hydrocephalus. It is therefore important to record all drainage volumes, as their absence can mean that the catheter is obstructed. It is also important to make sure that there is no catheter traction or CSF leakage [29]. | 1.00 | 1.00 |
| ■ | Leveling the system at the height of the external acoustic meatus allows the pressure transducer to be in line with the foramen of Monro. This action ensures the reliability of the monitoring and operation of the system. The leveling check should be performed at every change of patient's bedside height and at least once per shift [27,29,30]. | 1.00 | 1.00 |
| ■ | Early mobilization in patients with EVD is safe and feasible, and no major complications have been related to early mobilization [35,36]. Furthermore, it does not alter ICP and CPP parameters in patients with EVD [37] and, when in favorable clinical conditions (MAP > 80 mmHg, ICP < 20 mmHg and CPP > 70 mmHg), can be safely tolerated by the patient with minimal risk for adverse effects. It is important that a nurse and/or physiotherapist accompany the patient on the first time out of bed [37]. The zero point must be reviewed and maintained as prescribed by the medical team. In case of change of angulation or if the patient is sitting on the chair, the system must be reviewed. | 0.25 | 1.00 |

\* CVR—Content Validity Ratio. NA—Not applicable; EVD—external ventricular drain; CSF—cerebrospinal fluid; EAM—external acoustic meatus; ICP—intracranial pressure; CPP—cerebral perfusion pressure; MAP—mean arterial pressure.

Thus, 13 recommendations were reformulated and sent back to the participants in the second round. Nine recommendations achieved CVR ≥ 0.75, remaining unchanged after the reformulations made in the first round. Two recommendations showed CVR ≤ 0.75 and were reformulated based on the experts' suggestions. One recommendation showed RVC = 0.50 but remained unchanged as there were no additional suggestions.

Regarding the reformulated content, some highlights are safety of the EVD clamping for procedures, exams, and patient transport; periodic checking and leveling of the system; cerebrospinal fluid drainage flow; dressing, including periodicity of change and signs to be observed during the procedure; technique to check the permeability of the system; pain assessment concomitant to the Glasgow Coma Scale; technique for cerebrospinal fluid collection from the EVD system; and early mobilization when patient is stable.

As for the additional suggestions of the experts, P9 suggested the inclusion of recommendations for the transport of patients using EVD, considering that erroneous handling at this moment may make the system unviable and require its replacement. In response to the suggestion, a flowchart was formulated, describing step-by-step for preparation, transport, and return to the sector of origin, based on the recommendations of Stout et al. [38].

P8 suggested the inclusion of a recommendation about weaning off the EVD, since it is a temporary device. To meet the suggestion, a recommendation related to the description of gradual weaning techniques and immediate catheter closure was added. Nursing care related to the process was added, since the decision about the technique used is up to the neurosurgical team. A flowchart of mobilization of the EVD patient, adapted from Young et al. [35], was also included, in addition to suggestions made by participants in the first round.

## 4. Discussion

This study meets the need for a standardized clinical guideline for patients with EVD, especially considering that it was developed in an economically developing country, where available resources may vary in relation to those presented in current literature and practice.

When it comes to the evaluation process, we highlight the use of AGREE II as a well-established tool to assess methodological rigor and transparency in guideline development,

and its use is valuable and must be encouraged in clinical practice. Although it does not establish a minimum rate of agreement for approval (or not) of its domains, as highlighted in the updated version of the instrument published in 2017 [24], a cut-off point higher than 70% is being adopted by some authors [39–41].

In the present study, only the domains "stakeholder involvement" and "rigor of development" scored below 70% in the participants' evaluation. To address these points suggested by participants, multiprofessional members were included in the evaluation phase of the guideline and future contributions of allied health professional areas (physiotherapy, nutrition, psychology, for instance) are not excluded, and detailing of search strategies of the scoping review was added to the guideline, providing more clarity about methods.

Regarding changes related to the guideline's content, some points for discussion stand out: in-hospital transport, safe EVD clamping, and weaning off EVD.

The patient using EVD is often referred for imaging exams and surgical procedures. In-hospital transport by itself is a dangerous process for the patient [42], associated with the occurrence of adverse events and/or near misses, in addition to complications inherent to the patient's clinical status [34,43]. Planning of the transport is essential to prevent complications related to the procedure and should follow standardized routines and be conducted by a trained team [15,34,38].

A narrative review [12] about EVD publications, practices, and guidelines in the United States describes that 23 of the 30 hospitals included did not have a defined guideline for EVD clamping during transport. In addition, 66.7% always clamp the system for transport, and there are variations in ICP monitoring during the procedure (sometimes = 56.7%, always = 33.3%, never = 10%). This data shows the importance of including recommendations on in-hospital transport in this guideline, aiming to standardize actions and promote greater safety to care.

EVD clamping during transport and other procedures has been recommended [15,34] because of the risk of too much drainage, which may cause complications associated with intracranial hypotension, such as aneurysm rebleeding, subdural hematomas and even cerebral herniation [34]. However, risks of routine catheter clamping have been discussed, with emphasis on increased ICP, so that clamping should occur in a planned manner and consider the patient's clinical response [34].

The reflection about clamping is in line with suggestions proposed by the participants of this study, who stated that clamping is safe only when patient's clinical conditions are favorable, including controlled ICP (less than 20 mmHg), adequate cerebral perfusion pressure (CPP $\geq$ 70 mmHg), and stable hemodynamics (mean arterial pressure $\geq$ 70 mmHg). Such a recommendation applies both for hospital transport and for performing other procedures, such as changes in bedside position and bed bath, medication administration through the catheter, and early mobilization.

On the other hand, weaning off the EVD is still a point of discussion in literature [7,44–46], with no consensus on the superiority of the gradual weaning technique when compared to immediate catheter clamping. Gradual weaning occurs by increasing EVD height over the course of days, followed by catheter closure: if the patient does not present clinical worsening, increased ICP, and/or hydrocephalus during the period, the catheter is removed. Immediate EVD closure occurs as soon as the weaning process is started, following the same criteria of gradual weaning for catheter removal [44].

Given such divergence in the literature and the guideline's focus on EVD-related nursing care, the authors chose to cite both strategies and recommend rigorous neurological assessment of the patient. They emphasized the importance of paying attention to signs of neurological worsening and increased ICP (changes in level of consciousness, nausea/vomiting, visual changes and pupil pattern, and paresis/paresthesias) [47], regardless of the strategy used.

The recommendation about mobilization of the patient with EVD was changed after the evaluation process, including clinical conditions to be observed before starting the session. In addition to these criteria, the authors translated and adapted a flowchart already

available in the publications by Young et al. [34] and Moyer et al. [48] to be included in the guideline, aiming to support the multiprofessional teams' decision making when performing the procedure.

There are multiple complications related to immobilization during hospitalization [35], and early mobilization presents benefits related to respiratory and peripheral muscle strength improvement, better quality of life, and shorter mechanical ventilation time and delirium incidence, among others [35]. Despite the mobilization benefits already described, some factors hinder the neurocritical patient's mobilization, such as high fall risk, impulsiveness, increased ICP and reduced cerebral perfusion, and device displacement. Thus, strategies that allow better planning of the maneuver minimize the occurrence of complications, which justify changes in the guideline.

The main limitations of the study were as follows: a small sample size comprised of local professionals, which may lead to decisions that do not allow generalizations; a gap in literature when it comes to guidance to conduct studies on guidelines validity, not instruments only; pilot study was not carried out; and authors cannot assure that participants were not influenced by the other participants' opinions on the subject even if the questionnaire was carefully revised by the researchers in order to minimize bias. However, these limitations will be addressed during the implementation phase of this guideline, which might be re-evaluated and adapted to the current scenario.

## 5. Conclusions

This study provides an evidence-based clinical guideline for patients with EVD, enabling its use in clinical practice. The evaluation process followed two steps, and changes proposed were mainly related to a better detailing about its formulation and inclusion of information about transport, EVD weaning, safe catheter clamping, and patient mobilization. Adoption of evidence-based guidelines ensures safety of nursing care and strengthens the implementation of EBP in health institutions.

Although EVD is relatively common in critical care, few studies on the development and implementation of EVD guidelines are available in literature, especially for nurses. This paper aims to share the development and evaluation process of the proposed guideline, but further studies are needed to assess its impact on clinical practice when it comes to complications associated with EVD, duration of EVD placement, and mortality rate, and also to evaluate items that should be reformulated, added, or excluded from the guideline in accordance to the setting where it is being implemented. Cost-effectiveness studies are also suggested, especially considering lack of resources and financial crisis worldwide in consequence of 2019 Coronavirus Disease.

**Author Contributions:** Conceptualization, R.C.A.C., V.T.M.S., T.W.V. and C.R.B.; methodology, R.C.A.C., V.T.M.S. and T.W.V.; validation, R.C.A.C., C.R.B. and D.D.P.; formal analysis, V.T.M.S. and T.W.V.; investigation, V.T.M.S. and T.W.V.; resources, V.T.M.S. and T.W.V.; data curation, V.T.M.S. and T.W.V.; writing—original draft preparation, T.W.V., V.T.M.S. and B.R.A.; writing—review and editing, R.C.A.C., C.R.B. and D.D.P.; supervision, R.C.A.C., C.R.B. and D.D.P.; project administration, R.C.A.C., V.T.M.S. and T.W.V.; funding acquisition, T.W.V. and V.T.M.S. All authors have read and agreed to the published version of the manuscript.

**Funding:** This research received no external funding.

**Informed Consent Statement:** Informed consent was obtained from all subjects involved in the study.

**Data Availability Statement:** The data presented in this study are available on request from the corresponding author. The data are not publicly available because it is written in Portuguese, demanding translation for the corresponding language of interest.

**Conflicts of Interest:** The authors declare no conflict of interest.

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
