# Peer review of "External Ventricular Drains: Development and Evaluation of a Nursing Clinical Practice Guideline"

_nursrep, doi:10.3390/nursrep12040090_

Round 1
Reviewer 1 Report
Dear authors,
The manuscript addresses a relevant theme when considering the problem of external ventricular drains. However, it contains weaknesses, especially in the methodological field, which compromise the validity of the results, as well as the inferences of the conclusion.
Title – does not indicate the investigation of evidence of validity of the treated tool
Introduction – provides a good context for the issue of the latent variable. However, it is superficial in justifying the reason for developing a new tool.
Objective – not convergent to the type of study, which should not evaluate the “quality”, but the evidence of validity of this type of instrument, to confimr theorical/content validity.
Method – presents conceptual and procedural errors, resulting from the non-use of the main contemporary references in the field of psychometrics, for the development and investigation of evidence of instrument validity.
When proposing the type of study, one makes the mistake of calling it a methodological study. There is a conceptual mistake. It is a psychometric study.
The same is observed regarding the references for the development/construction of such contents, not being used or cited, which reflects in an absolutely theoretical construction, through literature and documents review, which leads to the absence of data from specialized practice, as well as the research experience and the users themselves.
Data analysis uses an inappropriate technique for the field, since the Delphi technique forces consensus among experts through controlled feedback. This technique has been identified as fragile since the 1970s for psychometric studies, due to the imposition of views and prejudices, reducing the possibility of divergent contributions; present deficient techniques for summarizing the answers, guaranteeing common interpretations; not exploit disagreement, ensuring intermediate consensus; include the judgment of a select group of people, which may not be representative; requires determined time and commitment from the participant. In addition, the Delphi technique, in its theory, requires four rounds of evaluation, which was not carried out.
Results and discussion – are not subject to analysis, due to methodological fragility, from construction to data analysis.
Conclusion is not supported, due to the aforementioned methodological aspects. In this context, it is not possible to conclude that the proposed.
Reviewer 2 Report
The present study about evidence-based clinical guideline for patients with EVD, is well written and conducted. The methods sound being the results quite informative and useful for nursing staff about patient with EVD safer mobilization/transportation and weaning off.
Author Response
We are grateful for the feedback provided on our research article. No further suggestions were made.